# Automatic Medical Report Generation Based on Cross-View Attention and Visual-Semantic Long Short Term Memorys

**DOI:** 10.3390/bioengineering10080966

**Published:** 2023-08-16

**Authors:** Yunchao Gu, Renyu Li, Xinliang Wang, Zhong Zhou

**Affiliations:** 1State Key Laboratory of Virtual Reality Technology and Systems, Beihang University, Beijing 100191, China; zy2106319@buaa.edu.cn (R.L.); wangxinliang@buaa.edu.cn (X.W.); zz@buaa.edu.cn (Z.Z.); 2Hangzhou Innovation Institute, Beihang University, Hangzhou 310051, China; 3Research Unit of Virtual Body and Virtual Surgery Technologies, Chinese Academy of Medical Sciences, 2019RU004, Beijing 100191, China

**Keywords:** automatic medical report generation, multi-view, Long Short Term Memorys

## Abstract

Automatic medical report generation based on deep learning can improve the efficiency of diagnosis and reduce costs. Although several automatic report generation algorithms have been proposed, there are still two main challenges in generating more detailed and accurate diagnostic reports: using multi-view images reasonably and integrating visual and semantic features of key lesions effectively. To overcome these challenges, we propose a novel automatic report generation approach. We first propose the Cross-View Attention Module to process and strengthen the multi-perspective features of medical images, using mean square error loss to unify the learning effect of fusing single-view and multi-view images. Then, we design the module Medical Visual-Semantic Long Short Term Memorys to integrate and record the visual and semantic temporal information of each diagnostic sentence, which enhances the multi-modal features to generate more accurate diagnostic sentences. Applied to the open-source Indiana University X-ray dataset, our model achieved an average improvement of 0.8% over the state-of-the-art (SOTA) model on six evaluation metrics. This demonstrates that our model is capable of generating more detailed and accurate diagnostic reports.

## 1. Introduction

Writing medical image diagnostic reports is time-consuming, laborious, and professionally demanding for radiologists. Thus, automatic medical report generation has become increasingly popular. This method can generate diagnostic text based on natural language for experienced radiologists, assist radiologists in completing their diagnosis, significantly reduce the burden of writing text, and accumulate diagnostic experience for radiologists or medical students who lack clinical experience.

Automatic generation of medical image diagnostic reports is based on traditional image captions [1,2,3,4,5], using an encoder–decoder framework [6]. Figure 1 is a report sample in the Indiana University (IU) X-ray dataset, which is widely used in automatic medical report generation. A sample is composed of multi-view images, findings, impressions, and Medical Text Indexer (MTI) tags. Findings and impressions are long diagnostic sentences with fixed sentence patterns. MTI tags are key words generated from the diagnostic sentences. Automatic medical report generation based on the characteristics of medical data has become a hot research topic in recent years. Zhang et al. [7] used knowledge maps and prior medical knowledge to enhance the features extracted from images. Xue et al. [8] used hierarchical Long Short Term Memory (LSTM) to generate long diagnostic sentences. Li et al. [9] used a template method to generate diagnostic sentences with fixed sentence patterns. Jing et al. [10] embedded the MTI tags predicted by the encoder in order to generate diagnostic reports. Similarly, Yuan et al. [11] extracted normalized medical concepts like MTI tags from the diagnostic report.

These studies have made progress in producing specific diagnostic reports; however, there are still two challenges in making full use of the characteristics of medical data to improve diagnostic report generation. The first challenge is the reasonable use of multi-perspective medical images. For example, there are two chest X-ray images in the report sample in Figure 1, so radiologists need to comprehensively evaluate the lesions from both images in the process of medical image diagnosis in order to write an accurate and comprehensive diagnostic report. The second challenge lies in the effective combination of multi-modal data because radiologists need to synthesize the observed image features and the key lesion features when writing the diagnostic report.

To overcome these two challenges, we propose a novel automatic report generation approach, which consists of two modules: Cross-View Attention Module and Medical Visual-Semantic LSTMs (CVAM+MVSL). We first develop CVAM based on the characteristics of multi-view medical images. The encoder receives the input of frontal and lateral chest X-ray images and outputs feature maps. Then, the feature maps are sent into two branches of CVAM. One is the single-view branch, which retains the view features, and the other is the cross-view branch, which integrates the multi-view features. The binary cross-entropy (BCE) loss function is used as the classification loss function, and mean square error (MSE) loss is used to unify the prediction results of the two branches. Next, we propose the module of MVSL to fuse visual features of the images and semantic features of lesions. The input for this structure is the multi-view image feature map given by the encoder and the embedding of predicted medical concepts. MVSL uses three LSTMs to process the multi-view image features and medical concepts, and the hidden layers of the LSTMs are utilized to determine the image area and medical concepts that should be examined at the moment. The medical visual-semantic features calculated by the fully connected layer are used as the input of the Sentence LSTM–Word LSTM to generate the diagnostic sentence.

The main contributions of our work are as follows:We propose CVAM to process multi-view medical images, which not only maintains the features of images but also makes full use of the complementary information of frontal and lateral chest X-ray images.We present MVSL to couple the visual features of images and the semantic features of lesions and to employ the hidden layers of LSTMs to determine the important features at the current moment.We perform extensive experiments and a user study to verify the effect and utility of the proposed methods. Results show that the proposed CVAM can significantly increase the area under curve (AUC) on both Chexpert and IU X-ray datasets. Compared with the previous methods, CVAM+MVSL can generate better medical reports with more information and higher accuracy.

## 2. Related Work

In this section, we introduce the related work on the topic of automatic medical report generation. The existing report generation methods mainly improve on encoders and decoders, but due to the entanglement of these two components, it is difficult to distinguish the key developments in these two components. Therefore, we introduce the relevant work in encoders and decoders separately.

Medical image analysis and processing based on deep learning play an increasingly important role in medical and health auxiliary diagnosis [12,13,14]. A key application of this technology is the automatic generation of medical images, which has received extensive attention in recent years [10,15,16,17,18,19]. Compared to other medical image analysis and processing tasks, the automatic generation of medical images is more challenging because it requires the modeling of both images and texts. So far, most of the current methods of automatic medical report generation are based on the framework of encoder–decoder [6,20,21,22,23,24,25].

The encoder is responsible for extracting image features. Jing et al. [10] used Convolutional Neural Networks (CNNs) to extract features from single-view chest X-ray images, taking out the results of the last layer of convolution as the feature expression of medical images. The majority of studies utilize CNNs as encoders and Recurrent Neural Network (RNNs) as decoders for report generation. However, Alahmadi et al. [21] employed RNNs as image encoders, adhering to the encoder–decoder machine translation model paradigm for caption generation. Li et al. [9] worked on multiple graphs to model the data structure and the transformation rules among different graphs. Zhang et al. [7] used the chest abnormality graph with prior knowledge to characterize image features. Yuan et al. [11] first proposed using two CNNs to process multi-view images; however, the utilization of multi-view medical images can be further improved. On the basis of these studies, our CVAM uses the characteristics of multi-view medical images to fuse and process multi-view medical image features.

The decoder receives the image features extracted by the encoder and generates the diagnostic sentence using LSTM. Xue et al. [8] proposed to use a Sentence LSTM–Word LSTM framework similar to [26] to generate multiple diagnostic sentences, which is widely used in the literature, such as [7,10,11]. Harzig et al. [24] contended that distinct patterns and data distributions between normal and abnormal cases can lead to biases in models. They addressed this by employing two separate word LSTMs to differentiate between the generation of abnormal and normal reports. Jing et al. and Yuan et al. [10,11] proposed introducing semantic features of lesions in the process of generating reports because these features can provide more abnormal information. Although we use the same basic Sentence LSTM–Word LSTM framework of the previous studies, the proposed MVSL pays more attention to how to effectively combine the visual features of medical images with the semantic features of lesions and provides the Sentence LSTM–Word LSTM with the data of image areas and lesion semantics when generating diagnostic sentences.

## 3. Materials and Methods

### 3.1. Datasets

The first dataset we use is IU X-ray [27], which contains 3959 medical diagnostic reports. Each report is labeled with chest X-ray images, impressions, findings, and MTI tags. We filter out samples of single-view images according to the experimental requirements. Following the conventions of the field of natural language processing, samples with fewer than 3 diagnostic sentences are removed, resulting in a total of 3331 samples. We pre-process the report text by converting it to lowercase text and replacing the words whose frequency is less than 3 with the $\langle unknown\rangle$ token. The filtered 1185 words constitute more than 99% of the word occurrence rate in the corpus. There are 155 independent tags in the MTI annotation of the original dataset, which is thus regarded as multi-label classification annotation. We randomly select 2000 samples for training, 678 samples for validation, and 653 samples for testing. The second dataset is Chexpert [28], in which a total of 224,316 chest X-ray images were collected and 14 common radiographic observations were labeled. From the entire dataset, 19,811 pairs of data with frontal and lateral images are utilized for training, 6619 pairs for validation, and 6608 pairs for testing. The purpose of using this dataset is to pre-train the encoder so that the model can extract effective medical image features. We then fine-tune the model on the IU X-ray dataset.

### 3.2. Methods Overview

In Figure 2, we use CVAM to process the features of multi-view medical images and to predict the tags that represent medical concepts contained in multi-view images as the semantic features of key lesions. The MVSL developed in this study receives image features and semantic features of lesions and generates a medical visual-semantic feature representing the sentence through the joint action of the two features. Using the characteristics of LSTMs, the module also records historical information to ensure information independence between diagnostic sentences. We then use the Sentence LSTM–Word LSTM to generate diagnostic reports.

### 3.3. Cross-View Attention Module (CVAM)

As shown in Figure 2, we use two CNNs to extract the features of the frontal-view image and lateral-view image, respectively. The last convolutional layer yields the feature map $V_{f}$ and $V_{l}$ ({$V_{f}$, $V_{l}$} ∈$R^{N\times D}$, where *N* is the W×H of the feature map and *D* is the depth). Each feature map then enters two branches. One is a single-view branch, where the multi-classification predictions $y_{f}$ and $y_{l}$ of *M* medical concepts are obtained by the fully connected layer. The other is a cross-view branch, as shown in Figure 3; $V_{f}$ and $V_{l}$ use SE-Attention [29] to enhance different lesion features channel-wise and then employ the following formula to complete the cross-view attention:(1)$$V_{af}=\lambda f\left(V_{f}\right)+\left(1-\lambda \right)f\left(V_{l}\right),$$
(2)$$V_{al}=\lambda f\left(V_{l}\right)+\left(1-\lambda \right)f\left(V_{f}\right),$$
where *f* is the SE-Attention and λ is a hyperparameter of [0.5, 1], which represents how many visual features of images are retained. By introducing (1−λ) visual features from complementary perspectives, $V_{\mathrm{af}}$ and $V_{\mathrm{al}}$ can be calculated by a fully connected layer to obtain the multi-classification predictions $y_{\mathrm{af}}$ and $y_{\mathrm{al}}$ of *M* medical concepts.

We then use MSE loss to unify the learning results of the single-view and cross-view branches. The loss function is shown below:(3)$$L_{\mathrm{single}-\mathrm{view}}=\sum_{i=1}^{M}{\left({y_{f}}_{i}-{y_{l}}_{i}\right)}^{2},$$
(4)$$L_{\mathrm{cross}-\mathrm{view}}=\sum_{i=1}^{M}{\left({y_{\mathrm{af}}}_{i}-{y_{\mathrm{al}}}_{i}\right)}^{2},$$
(5)$$L_{\mathrm{CVAM}}=L_{\mathrm{single}-\mathrm{view}}+L_{\mathrm{cross}-\mathrm{view}}.$$

### 3.4. Medical Visual-Semantic LSTMs (MVSL)

Visual features of images include information about objects and locations, and medical concepts can be directly used as the semantic information for key lesions. The fusion of these features can produce a diagnostic sentence that includes the location and type of the disease. In Figure 2, the MVSL receives the visual features of medical images and the semantic features of lesions from CVAM. The visual features of different perspectives and branches are integrated into a visual feature vector through the fully connected layer, and the semantic features of the lesions are embedded into a semantic feature vector. As shown in Figure 4, three LSTMs handle the visual feature vectors and semantic feature vectors, which can be defined as:(6)$$h_{s}^{F}={\mathrm{LSTM}}^{F}\left(F,h_{s-1}^{F}\right),$$
where *s* refers to the diagnostic sentence that is currently being generated. *F* ∈ {$V_{f}$,$V_{l}$,MS} represents variables related to the frontal view, lateral view, or medical semantic feature. Then, three hidden layers are used to calculate the visual vector attention ($a_{s}^{V_{f}}$, $a_{s}^{V_{l}}$) and semantic vector attention ($a_{s}^{\mathrm{MS}}$):(7)$$a_{s}^{F}=\mathrm{softmax}\left(W_{a}^{F}\tanh\left(W_{F,a}^{F}F+W_{h_{F},a}^{F}h_{s}^{F}\right)\right),$$
where softmax(·) is the function of softmax layer and $W_{a}^{F}$, $W_{F,a}^{F}$, $W_{h_{F},a}^{F}$ are parameter matrices. Then, the visual and semantic attention vectors are obtained by the following formulas:(8)$${V_{f}^{\mathrm{att}}}_{s}=\sum_{i=1}^{N}{a_{s}^{V_{f}}}_{i}{V_{f}}_{i},$$
(9)$${V_{l}^{\mathrm{att}}}_{s}=\sum_{i=1}^{N}{a_{s}^{V_{l}}}_{i}{V_{l}}_{i},$$
(10)$${{\mathrm{MS}}^{\mathrm{att}}}_{s}=\sum_{i=1}^{M}{a_{s}^{\mathrm{MS}}}_{i}{\mathrm{MS}}_{i}.$$

With a fully connected layer *W*, the three vectors are then integrated to obtain the medical visual-semantic feature (MVS):(11)$${\mathrm{MVS}}_{s}=W\left({V_{f}^{\mathrm{att}}}_{s}+{V_{l}^{\mathrm{att}}}_{s}+{{\mathrm{MS}}^{\mathrm{att}}}_{s}\right).$$

### 3.5. Sentence LSTM–Word LSTM

The Sentence LSTM generates the topic of the current diagnostic sentence and the control vector of whether to continue to generate the diagnostic sentence. The topic of this sentence is generated by the hidden layer in the Sentence LSTM and medical visual-semantic feature:(12)$${\mathrm{dtopic}}_{s}=\tanh\left(W_{s,\mathrm{sent}}h_{s}^{\mathrm{sent}}+W_{s,\mathrm{MVS}}{\mathrm{MVS}}_{s}\right),$$
where *s* is the sequence number of the sentence being generated and $W_{s,\mathrm{sent}}$ and $W_{s,\mathrm{MVS}}$ are weight parameters. The control vector is generated with the current *h* and the previous *h* of the LSTM:(13)$${\mathrm{stop}}_{s}=W_{\mathrm{stop}}\tanh\left(W_{\mathrm{dtopic},s-1}h_{s-1}^{\mathrm{sent}}+W_{\mathrm{dtopic},s}h_{s}^{\mathrm{sent}}\right),$$
where $W_{\mathrm{stop}}$, $W_{\mathrm{dtopic},s-1}$, $W_{\mathrm{dtopic},s}$ are weight parameters.

The Word LSTM uses the topic and embedding vectors to continuously calculate and output words:(14)$$h_{t}^{\mathrm{word}}={\mathrm{LSTM}}^{\mathrm{word}}\left(x_{t},{\mathrm{dtopic}}_{s},h_{t-1}^{\mathrm{word}}\right),$$
where *t* is the *t*-th word being generated in one sentence and $x_{t}$ is the embedding vector of the input. Then, the *h* of each time is used to complete the word generation:(15)$$p\left(\mathrm{word}|h_{\mathrm{word}}\right)=W_{\mathrm{output}}\left(W_{\mathrm{word}}h_{\mathrm{word}}\right),$$
where $W_{\mathrm{output}}$, $W_{\mathrm{word}}$ are weight parameters.

### 3.6. Training Loss

The input of our model includes: (1) frontal and lateral images $I_{f}$ and $I_{l}$; (2) tag *T* for medical concepts; (3) a diagnostic report with *s* sentences; (4) stop signal *P*.

The model extracts image features from $I_{f}$ and $I_{l}$ by CNN. Then, the features enter the two branches, and the multi-label classification is conducted. BCE loss is used to calculate the predicted value and the loss of *T*:(16)$$L_{\mathrm{BCE}}=-\sum_{j\in \{f,l,\mathrm{af},\mathrm{al}\}}\sum_{i=1}^{M}T_{i}\log p_{j,i}+\left(1-T_{i}\right)\log\left(1-p_{j,i}\right).$$

With the $L_{\mathrm{CVAM}}$, the loss of the encoder is formulated as:(17)$$L_{\mathrm{encoder}}=\alpha L_{\mathrm{BCE}}+\beta L_{\mathrm{CVAM}}.$$

Next, the Sentence LSTM generates the medical topic vector and control vector *s* times and utilizes cross-entropy loss to calculate the loss of the control vector:(18)$$L_{\mathrm{stop}}=-\sum_{i=1}^{S}y_{i}\log\left(p_{i}\right).$$

Finally, in the Word LSTM, we use cross-entropy to calculate the loss of words and ground truth of *S* sentences:(19)$$L_{\mathrm{word}}=-\sum_{i=1}^{S}\sum_{j=1}^{\mathrm{len}(s_{i})}y_{i,j}\log\left(p_{i,j}\right).$$

Combining all the loss described above yields the total training loss:(20)$$L_{\mathrm{total}}=\lambda_{e}L_{\mathrm{encoder}}+\lambda_{s}L_{\mathrm{stop}}+\lambda_{w}L_{\mathrm{word}}.$$

## 4. Results

### 4.1. Evaluation Metrics

We use AUC as the evaluation metric for encoder training. AUC is a numerical value ranging from 0 to 1, representing the area under the receiver operating characteristic curve, which illustrates the relationship between the model’s true positive rate and false positive rate.

To evaluate medical report generation, we use standard image caption evaluation metrics BLEU [30], ROUGR [31], and CIDER [32]. BLEU is an evaluation metric based on n-grams, where n can be 1, 2, 3, and so on. It assesses the degree of n-gram overlap between the generated text and multiple reference texts, serving to evaluate the quality of generated text that corresponds to precision in classification tasks. We use the ROUGE-L metric from the ROUGE metric family to assess the similarity between the texts in this paper. ROUGE-L is an evaluation metric based on the longest common subsequence. It calculates the length of the longest common subsequence between the generated text and the reference text to measure their similarity. CIDER is a fusion of BLEU and the vector space model. It treats each sentence as a document and calculates the cosine angle of TF-IDF vectors (with terms being n-grams instead of words), obtaining the similarity between candidate and reference sentences. Additionally, CIDER leverages inverse document frequency, enhancing the significance of pivotal vocabulary terms within the corpus.

### 4.2. Baselines

#### 4.2.1. Multi-Label Classification

The first baseline is a multi-task learning model for processing the input of frontal and lateral chest radiographs. The second comparative model adds MSE loss to the multi-task learning model used in [11]. Because of the design of single-view and cross-view in CVAM, we perform comparative experiments to verify the effectiveness of using two branches. We also investigate the effect of only using ImageNet during the fine-tuning on the IU X-ray dataset and compare this effect with those of the other models.

#### 4.2.2. Medical Report Generation

We compare our method with several basic models of image caption and the state-of-the-art methods for automatic medical report generation. For the method TieNet [16], CARG [33], and SentSAT+KG [7], we compare them with the results reported in [7]. In addition, we reproduce HLSTM [26] and CoAtt [10] (the original method is changed to frontal and lateral input) for comparison. Our baseline can be regarded as a type of encoder (pre-trained on Chexpert) + Sentence LSTM + Word LSTM. Our CVAM adds CVAM to integrate multi-perspective features based on the baseline. Our CVAM+MVSL is a method for verifying the effectiveness of the proposed method.

### 4.3. Implementation Details

We employ ResNet50 [34] and use the size of 256 × 256 to train the encoder. The parameter λ in CVAM is 0.6 according to the experimental results shown in Table 1. Then, the 8 × 8 × 2048 feature map is output by the last convolutional layer as medical image features. The other output of the encoder is the prediction of 155 medical concept tags. After the softmax layer, 10 prediction tags with the highest probability are selected as the semantic features of the key lesions, which are expressed by 512-dimensional word embedding. In the decoder, the dimensions of the hidden state and word embedding are set to 512. We set the Sentence LSTM to generate 6 sentences for each sample during training. Each sentence retains the first 30 words processed and uses the $\langle pad\rangle$ token to fill in when a sentence contains fewer than 30 words.

We use the Adam optimizer for parameter learning. The learning rates of the encoder, MVSL, and Sentence LSTM–Word LSTM are set to $1\times 10^{-3}$, $1\times 10^{-4}$, and $1\times 10^{-4}$, respectively. After training the encoder on Chexpert, we fine-tune it on the IU X-ray dataset. The loss of the encoder includes BCE loss of multi-label classification and MSE loss of CVAM. The loss of the decoder includes the cross-entropy loss of the stop control variable in the Sentence LSTM and words in the Word LSTM. In the experiment, we set α and β in $L_{encoder}$ to 1 and 0.05, respectively. In $L_{total}$, the values of $\lambda_{e}$, $\lambda_{s}$, and $\lambda_{w}$ are 1.

### 4.4. Quantitative Analysis

#### 4.4.1. Multi-Label Classification

The AUC of multi-label classification is shown in Table 1. As can be seen in the table, the CVAM structure can achieve excellent results on both datasets. On the Chexpert dataset, our model performs 3.7% better than the baseline Multitask. A gap exists between the classification performance of the baseline for frontal and lateral medical images, but our method can effectively make up for this gap. Our naive model only uses the cross-view branch, and the performance of the model in processing frontal images is slightly reduced. When the two branches are utilized, our model can achieve the best performance. When fine-tuned on the IU X-ray dataset, the model using CVAM performs 3.1% better than Multitask–ImageNet. These results show that adding CVAM in the feature extraction stage can effectively process multi-view medical images, improve classification performance while making up for the performance gap of multi-view images, and obtain more useful lesion features for diagnosis.

#### 4.4.2. Medical Report Generation

Table 2 shows the results of the generated report. Note that because some methods do not release the source code or lack important details, we compare the results reported in [7]. The methods in the middle part of Table 1 are reproduced with the same data partition as our method. Compared with the previous work, our CVAM+MVSL method can achieve better performance on BLEU-n and CIDER, although the comparison with TieNet, CARG, and SentSAT+KG may be unfair due to different experimental settings. In the reproduced HLSTM and CoAtt, we change the input to multi-view medical images in order to make a more reasonable comparison. Our CVAM+MVSL shows better performance on BLEU-n and CIDER, indicating that the diagnostic report generated by our method has a higher degree of lexical overlap with the real diagnostic report. CIDER utilizes reverse document frequency, which increases the significance of important words in the corpus. The results show that our method can effectively predict important words in the medical corpus. In ROUGE, the effect of our method is slightly reduced compared with that in CoAtt. This might be because ROUGE is a metric that considers the recall rate, which can provide more information for the report generated by our method.

### 4.5. Qualitative Analysis

As shown in Figure 5, we randomly selected the visual results of automatically generated medical diagnostic reports, along with the visualization of text and visual-semantic attention. As can be seen from the figure, our CVAM+MVSL method possesses the following characteristics. Firstly, CVAM enables the model to extract more effective lesion features and predict lesions more accurately. In our model, the report generated for the first sample accurately described the symptom of “cardiomegaly”, while the report for the second sample correctly depicted “scarring” and “emphysema”, which CoAtt failed to recognize. We believe that this improvement is due to the utilization of three-dimensional information provided by both frontal and lateral views. Secondly, MVSL iteratively selects different visual and semantic features to generate more diverse diagnostic reports. For the third sample without diseases, our method generates valuable descriptions beyond conventional descriptions, such as the diagnosis of “cardiac mediastinal contour and pulmonary vascular system”. Finally, our method can make full use of the joint and complementary information of multi-view medical images to complete the diagnosis. The features of “cardiac hypertrophy” are captured in the frontal and lateral view of the second sample. When the third sample is generated to describe the “visible bone structure of the chest”, the lateral image, instead of the frontal image, captures the bone structure characteristics.

### 4.6. Clinical Validation

We invited six radiologists to evaluate the diagnostic reports generated by our method and designed a questionnaire with five questions concerning “fluency of generated reports”, “comprehensiveness of generated reports”, “accuracy of generated reports in describing diseased sites”, “accuracy of attention map in capturing diseased sites”, and “utility of automatic generation of diagnostic reports”. The response to each question was set with a score ranging from 0 to 10, with 0 being the worst and 10 the most satisfactory. As Table 3 shows, radiologists gave a high evaluation to the fluency of the generated report and the effect of the algorithm. The lowest score is for “comprehensiveness of report generation”. This can be attributed to the limitation of the IU X-ray dataset, as the model cannot effectively generate the words outside the dataset well. However, this problem can be alleviated by using more diagnostic texts to pre-train the decoder. The scores of the third and fourth questions are not bad; we think that the encoder focuses on the discriminant features when extracting features and thus cannot effectively capture all the key features. This can be solved by using methods proposed by [35]. The average ICC [36] value of the user study is 0.966, which confirms the consistency and reliability of our results (greater than 0.75).

## 5. Conclusions

We propose a novel automatic report generation approach composed of Cross-View Attention Module (CVAM) and Medical Visual-Semantic LSTMs (MVSL). The CVAM integrates multi-view medical image features to provide more effective visual features of images and semantic features of key lesions. The MVSL can integrate visual and semantic data to provide discriminative information for diagnostic report generation. Applied to the open-source IU X-ray dataset, our model achieved an average improvement of 0.8% over the state of the art (SOTA) on six evaluation metrics. This demonstrates that our model is capable of generating more detailed and accurate diagnostic reports. We attribute this improvement to the introduction of multi-perspective information, which enables the model to focus on pathological changes that cannot be captured by a single perspective. In the future, we plan to use the weakly supervised localization method to extract more discriminative features from the encoder and use more medical texts to pre-train the decoder so that the model can generate more varied sentences.

## Figures and Tables

**Figure 1 bioengineering-10-00966-f001:**
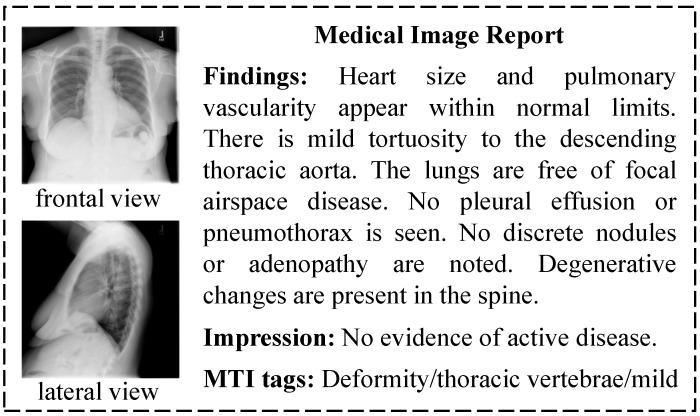
A diagnostic report with multi-view chest X-ray images.

**Figure 2 bioengineering-10-00966-f002:**
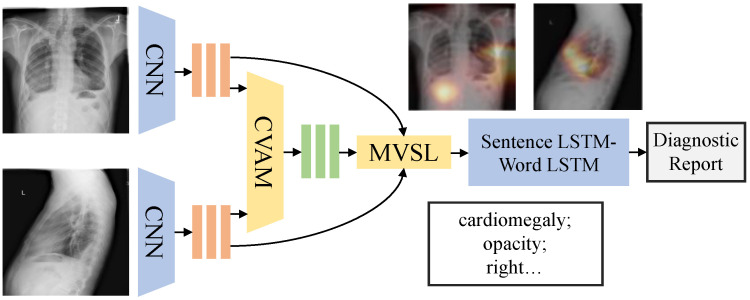
Illustration of the methods. The orange rectangle represents the visual features of images; the green rectangle represents the semantic features of key lesions. CVAM is our proposed Cross-View Attention Module for processing multi-view medical images, and MVSL is our designed Medical Visual-Semantic LSTMs integrating visual and semantic features.

**Figure 3 bioengineering-10-00966-f003:**
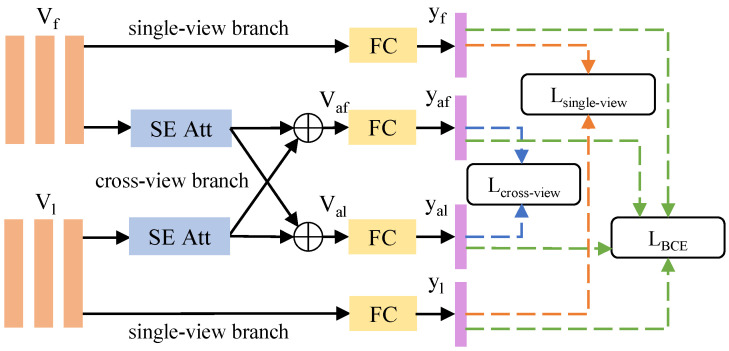
The architecture of the Cross-View Attention Module.

**Figure 4 bioengineering-10-00966-f004:**
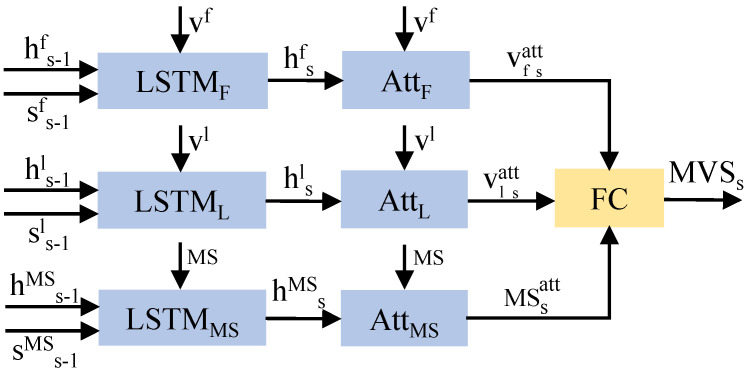
The architecture of Medical Visual-Semantic LSTMs.

**Figure 5 bioengineering-10-00966-f005:**
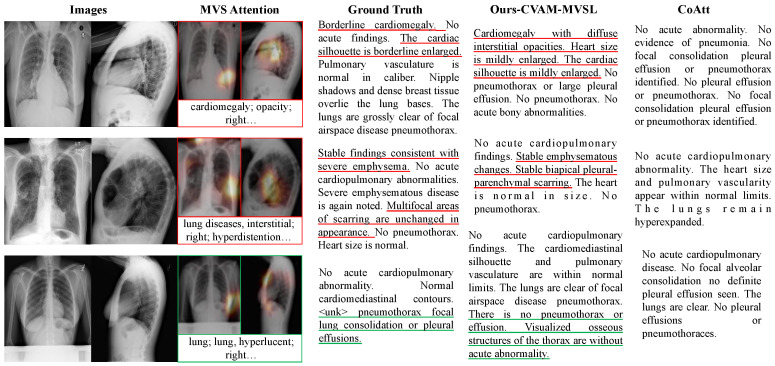
Medical image diagnostic reports generated from the IU X-ray dataset, multi-view images, and focused semantic attention. The underlined text indicates the correspondence between our method and ground truth. The red bounding boxes or underlines refer to the diagnosis of the diseased part, and the green indicates that no disease is found. Three semantic features of lesions are displayed here.

**Table 1 bioengineering-10-00966-t001:** The AUC of frontal view, lateral view, and total of baseline and CVAM at different λ. The upper and lower parts are the experimental results of pre-training on the Chexpert and fine-tuning on the IU X-ray datasets, respectively. Bold indicates the best result.

Datasets	Methods	AUC-f	AUC-l	AUC
Chexpert	Multitask	0.822	0.800	0.810
	Loss	0.835	0.829	0.832
	Ours-naive	0.830	0.829	0.830
	Ours (λ = 0.50)	0.838	0.834	0.836
	Ours (λ = 0.60)	0.842	**0.838**	**0.840**
	Ours (λ = 0.75)	0.838	0.834	0.836
	Ours (λ = 0.90)	**0.843**	0.837	**0.840**
IU X-ray	Multitask–ImageNet	0.874	0.864	0.869
	Multitask	0.879	0.872	0.875
	Loss	0.890	0.886	0.889
	Ours (λ = 0.50)	0.896	0.893	0.895
	Ours (λ = 0.60)	**0.897**	**0.894**	**0.896**
	Ours (λ = 0.75)	0.892	0.891	0.891
	Ours (λ = 0.90)	0.890	0.890	0.890

**Table 2 bioengineering-10-00966-t002:** Evaluation results for the IU X-ray dataset compared with previous methods. The top part shows results reported in [7], the middle part results we reproduce, and the bottom part the results of our methods. Bold indicates the best result.

Methods	BLEU-1	BLEU-2	BLEU-3	BLEU-4	CIDER	ROUGE
TieNet [16]	0.330	0.194	0.124	0.081	-	0.311
CARG [33]	0.359	0.237	0.164	0.113	-	0.354
SentSAT+KG [7]	0.441	0.291	0.203	0.147	0.304	0.367
HLSTM [26]	0.432	0.271	0.188	0.137	0.310	0.377
CoAtt [10]	0.441	0.284	0.199	0.147	0.397	**0.391**
Ours-baseline	0.442	0.284	0.201	0.148	0.349	0.373
Ours-CVAM	0.455	0.289	0.204	0.150	0.392	0.384
Ours-CVAM+MVSL	**0.460**	**0.294**	**0.207**	**0.152**	**0.409**	0.385

**Table 3 bioengineering-10-00966-t003:** Statistics for the questionnaire in the user study.

Questions	Mean	Standard Deviation
1	8.33	0.52
2	7.67	0.52
3	8.17	0.41
4	8.17	0.41
5	10.00	0.00

## Data Availability

Not applicable.

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
