# Peer review of "Automatic Medical Report Generation Based on Cross-View Attention and Visual-Semantic Long Short Term Memorys"

_bioengineering, 2023, doi:10.3390/bioengineering10080966_

Round 1

Reviewer 1 Report

This paper proposes an automatic medical report generation system for multi-view chest X-ray data, with two main features: firstly, a Cross-View Attention Module (CVAM) is designed to integrate information from two different X-ray views (frontal/lateral). Secondly, a Medical Visual-Semantic LSTM (MVSL) is formulated as a combination of three separate LSTMs, which process variables related to the image frontal view, image lateral view and semantic features respectively. Experiments on two public datasets suggests improved AUC for multi-label classification, and improved medical report quality by qualitative analysis.

Some issues might be considered:

1. While the motivation for using multiple views is sound, it might be appropriate to provide examples where the use of multiple views allows insights/diagnoses that a single view does not, if possible.

2. Model behaviour when the multiple view input is inconsistent (e.g. the two X-rays are mistakenly taken from different patients), might be briefly discussed - would the model be able to detect an error, or will it simply proceed under the assumption that the views are consistent, even when clearly not?

3. In Section 3.3, Equations 1 & 2 appear to indicate that the V_af and V_al cross-view feature vectors, are simply linear combinations of features derived from different CNNs. In theory, the elements of the two independent feature vectors (Vf and Vl) may not meaningfully correspond to each other. As such, is there any good justification for combining the features this way? This might be discussed.

4. In general, there is insufficient ablation experimentation performed, to support the proposed extensions. For example, for the CVAM, Ours-naive in Table 1 uses only the cross-view branch without adjusting the lambda parameter, and the baseline model without cross-view branch does not appear to be attempted (unless it is the Loss model). Similarly, different combinations of the three LSTMs in MVSL were not explored, to gauge their individual contributions.

5. For the qualitative analysis in Section 4.5, it might be clarified if the examples were randomly chosen, or picked.

N/A

Author Response

Q1: While the motivation for using multiple views is sound, it might be appropriate to provide examples where the use of multiple views allows insights/diagnoses that a single view does not, if possible.

A1: In fact, we demonstrated such examples in the qualitative evaluation. For instance, the second example included emphysema and multifocal scar areas, which require dual views to comprehensively determine the lesion type and severity. Therefore, our model correctly identified both of these lesions, whereas the single-view CoAtt couldn't recognize these two types of lesions. The corresponding descriptions have been updated in the quantitative experiments.

Q2: Model behaviour when the multiple view input is inconsistent (e.g. the two X-rays are mistakenly taken from different patients), might be briefly discussed - would the model be able to detect an error, or will it simply proceed under the assumption that the views are consistent, even when clearly not?

A2: Yes, we assume that the two views remain consistent. Since assessing the consistency between the two views is different from the task of report generation, we consider measuring the consistency between the two views to be a design aspect beyond the algorithm itself.

Q3: In Section 3.3, Equations 1 & 2 appear to indicate that the V_af and V_al cross-view feature vectors, are simply linear combinations of features derived from different CNNs. In theory, the elements of the two independent feature vectors (Vf and Vl) may not meaningfully correspond to each other. As such, is there any good justification for combining the features this way? This might be discussed.

A3: Yes, V_af and V_al are simple linear combinations of features from two views and do not have explicit meanings. We did this to fuse the information from both views in order to obtain features that cannot be acquired from a single view alone. For example, certain lesions may require the combination of information from both views to make accurate assessments.

Q4: In general, there is insufficient ablation experimentation performed, to support the proposed extensions. For example, for the CVAM, Ours-naive in Table 1 uses only the cross-view branch without adjusting the lambda parameter, and the baseline model without cross-view branch does not appear to be attempted (unless it is the Loss model). Similarly, different combinations of the three LSTMs in MVSL were not explored, to gauge their individual contributions.

A4: For CVAM, indeed, we did not conduct experiments with a single-branch baseline model. This is because the disease labels correspond to combined labels from both views, and a single-branch model cannot find the corresponding labels. The reason for not exploring different combinations of LSTMs in MVSL is also the same.

Q5: For the qualitative analysis in Section 4.5, it might be clarified if the examples were randomly chosen, or picked.

A5: The examples were randomly selected, and the corresponding descriptions have been updated in Section 4.5.

Reviewer 2 Report

In the Abstract, on lines 12-13 very subjective statements are written. Authors must justify what they write.
(lines 17, 22) no matter how good the authors' intentions are, physicians never accept being replaced by a machine. Consequently, they also do not accept being replaced in the task of image analysis and writing a report, even an artificial intelligence system is not 100% correct and the failures of an automatic system can lead to the pathology not being correctly identified and, in an extreme case, if the patient is sick or very sick, it may contribute to the patient's death. That is, the authors must choose the correct words, not mention that they will replace a physician, but that they will assist the physician in the diagnosis and assist in the preparation of a medical report.
In the paragraph on lines 22-34 it is not correct to start a sentence with a reference; the authors must write the name of the author and only then write the corresponding reference. This situation also occurs in chapter 2.
Figure 2 is on line 73 and is only cited on line 122. All figures must be close to the text where they are cited.
Chapter 2 “Related Word” has 3 sections where each section is a paragraph. It doesn't make sense for a section to be a paragraph. Either more text is added to each section to be at least two paragraphs long, or subsection headings are removed.
Chapter 2 very confusing: (i) it is not clear the logic of presenting subsections 2.1, 2.2 and 2.3, as apparently it makes more sense to be the contents of 2.2, 2.3 and 2.1; (ii) at the beginning of this chapter there should be a paragraph explaining what is described in this chapter, mentioning the sequence of subsections and their interconnection; (iii) as a state of the art is poor because it does not include more detail of each of the methods described, nor does it includes which data are used in each of the papers, nor does it include which conclusions are reached by each of the papers.

Line 109: It doesn't make sense to discarded “diagnostic sentences less than 3” because the report can be just 3 big sentences that eventually correspond to 3 big paragraphs.
In chapter 3 it is not clear which part of the methodology was created by the authors and which part of the methodology was inspired / based on an existing methodology.
Avoid presenting Table 1 and Table 2 consecutively. Place tables next to the text when they are cited and ensure that there is text between the two tables. Also bear in mind that the caption of a table must be before the table and not after the table (see any MDPI scientific paper). In table 2 it is not shown in the first row which type of metric was used.
Line 238 “performs 3.1% better” depends on the metric used. 3.1% can be a big difference or an almost irrelevant difference.
In the results chapter, the authors should give emphasis to quantitative results, including the presentation of mathematical expressions of the metrics used.
Very poor conclusions, it is not written the quantitative results achieved in this work, nor are the subjective words (lines 303 and 304) used in the conclusions justified.
The references presented do not correspond to a complete bibliographical research work, as 75% of the references presented are of authors with Chinese names (which may suggest a breakdown of ethnicity). When searching in scientific search engines (Scopus, etc.) there are many authors from North America, Western and Central Europe who should also be included; authors are recommended to complement the state of the art with more works of excellent quality written by American and/or European authors.

Author Response

Q1: In the Abstract, on lines 12-13 very subjective statements are written. Authors must justify what they write.

A1: (line10-13) Thank you for your feedback. We have revised the subjective statements and added quantitative descriptions.

Q2: (lines 17, 22) no matter how good the authors' intentions are, physicians never accept being replaced by a machine. Consequently, they also do not accept being replaced in the task of image analysis and writing a report, even an artificial intelligence system is not 100% correct and the failures of an automatic system can lead to the pathology not being correctly identified and, in an extreme case, if the patient is sick or very sick, it may contribute to the patient's death. That is, the authors must choose the correct words, not mention that they will replace a physician, but that they will assist the physician in the diagnosis and assist in the preparation of a medical report.

A2: I agree with your perspective that doctors will never be replaced by machines. In our paper, we aim to emphasize that automatically generated medical image diagnostic reports can serve as a valuable reference for doctors, acting as a tool to reduce their workload, rather than replacing them. The ultimate diagnosis will still be determined by the medical professionals.

Q3: In the paragraph on lines 22-34 it is not correct to start a sentence with a reference; the authors must write the name of the author and only then write the corresponding reference. This situation also occurs in chapter 2.

A3: Thank you for your feedback. I have made revisions regarding this issue as suggested.

Q4: Figure 2 is on line 73 and is only cited on line 122. All figures must be close to the text where they are cited.

A4: Thank you for your feedback. I have made revisions regarding this issue as suggested.

Q5: Chapter 2 “Related Word” has 3 sections where each section is a paragraph. It doesn't make sense for a section to be a paragraph. Either more text is added to each section to be at least two paragraphs long, or subsection headings are removed.

A5: Answers refer to A6.

Q6: Chapter 2 very confusing: (i) it is not clear the logic of presenting subsections 2.1, 2.2 and 2.3, as apparently it makes more sense to be the contents of 2.2, 2.3 and 2.1; (ii) at the beginning of this chapter there should be a paragraph explaining what is described in this chapter, mentioning the sequence of subsections and their interconnection; (iii) as a state of the art is poor because it does not include more detail of each of the methods described, nor does it includes which data are used in each of the papers, nor does it include which conclusions are reached by each of the papers.

A6: Thank you for your feedback. We have added some sentences to the original text, enhancing the logical relationship between sub-sections. As the task of automatic report generation is based on an encoder-decoder architecture, previous methods have primarily focused on improving both the encoder and decoder. However, due to the intertwined nature of these two components, it becomes challenging to identify key technological innovations, such as single-view and multi-view encoders, as well as multi-layer LSTM structures and the introduction of semantic features in the decoder. Therefore, we ultimately decided to divide the content into two sub-sections, decoupling the discussion of the two parts, to clarify the logic behind our work.

Q7: Line 109: It doesn't make sense to discarded “diagnostic sentences less than 3” because the report can be just 3 big sentences that eventually correspond to 3 big paragraphs.

A7: This operation is the default operation in the Natural language processing direction, and other methods do the same.

Q8: Avoid presenting Table 1 and Table 2 consecutively. Place tables next to the text when they are cited and ensure that there is text between the two tables. Also bear in mind that the caption of a table must be before the table and not after the table (see any MDPI scientific paper). In table 2 it is not shown in the first row which type of metric was used.

A8: Thank you for your advice. I have made revisions regarding this issue as suggested. In Table 2, we have used six metrics, namely BLEU-1, BLEU-2, BLEU-3, BLEU-4, CIDER, and ROUGE. Due to the large number of metrics, detailed explanations were not provided in the caption.

Q9: Line 238 “performs 3.1% better” depends on the metric used. 3.1% can be a big difference or an almost irrelevant difference.

A9:

  1. Undoubtedly, a model with higher metrics performs better, and the 3.1% improvement quantifies this "better" performance.

  1. From the experiments on report generation based on this task, it is evident that multi-view processing has played a significant role, and this can be attributed to the 3.1% improvement in the metrics.

Q10: In the results chapter, the authors should give emphasis to quantitative results, including the presentation of mathematical expressions of the metrics used.

A10: Apart from AUC, the computation of other metrics is more complex and would require a significant amount of space. Additionally, these metrics are commonly used in natural language tasks. Therefore, we have decided not to present them in the Results section.

Q11: Very poor conclusions, it is not written the quantitative results achieved in this work, nor are the subjective words (lines 303 and 304) used in the conclusions justified.

A11: Thank you for your feedback. I have made revisions regarding this issue as suggested.

Q12: The references presented do not correspond to a complete bibliographical research work, as 75% of the references presented are of authors with Chinese names (which may suggest a breakdown of ethnicity). When searching in scientific search engines (Scopus, etc.) there are many authors from North America, Western and Central Europe who should also be included; authors are recommended to complement the state of the art with more works of excellent quality written by American and/or European authors.

A12: Your concern is very valid. However, we have not deliberately filtered the articles we cited. These papers were all published in top conferences and journals, which should be considered a coincidence.

Round 2

Reviewer 1 Report

We thank the authors for addressing our previous comments.

N/A

Author Response

Thank you for your valuable feedback.

Reviewer 2 Report

read attached file

Author Response

Q5': The problem persists: the 3 sections between lines 77 to 109 are just 3 paragraphs. Although this is not a very serious situation, it is recommended that any subsection has at least two paragraphs.

A5': We have merged subchapters 2.1, 2.2 and 2.3.

Q6': It was suggested to put a paragraph between chapter 2 and section 2.1, explaining what is presented in chapter 2. This paragraph should act as an introduction to chapter 2. The authors tried to answer this question in the "reply to reviewers" but failed to do not put this information on paper. An introductory paragraph is still missing in Chapter 2.

A6': We have added relevant descriptions in Chapter 2.

Q7': Reply from reviewer: include this justification on paper so that the reader understands this limitation.

A7': We have added relevant expressions in line 119-123.

Q10': a reader likes to know what metrics are used. When the expressions are very complex and are not shown, it is important for the reader to understand whether the metrics are linear, polynomial or other types of functions and also what the range of variation (generally varies between 0 and 1, but can vary between - 1 and +1). Again, although this information is not decisive in the evaluation of a scientific paper, it is always good to have it present.

A10': We have added relevant expressions in line 210-227.

Q11': For a 10-page work, it seems impoverishing to be summarized in approximately 12 lines. I recommend that the authors complement the conclusions, in addition to stating that the proposed model is better 0.8%, add information to justify what the proposed model has (and others do not have) so that the result is +0.8%.

A11': We have added relevant expressions in Chapter 5.

Q12': References used to create the state of the art are insufficient. Also inappropriate response to reviewer, since the authors did not follow the reviewer's recommendation. Authors must complement the state of the art with additional (at least five) international papers from North America and/or Western Europe and/or Central Europe. The authors may use these and/or other articles: Automatic generation of textual summaries from neonatal intensive care data By:Portet, F (Portet, Francois) [1] ; Reiter, E (Reiter, Ehud) [1] ; Gatt, A (Gatt, Albert) [1] ; Hunter, J (Hunter, Jim) [1] ; Sripada, S (Sripada, Somayajulu) [1] ; Freer, Y (Freer, Yvonne) [2] , [3] ; Sykes, C (Sykes, Cindy) 114 citations A novel colonoscopy reporting system enabling quality assurance van Doorn, SC; van Vliet, J; (...); Dekker, E 29 citations

A12': We have added seven corresponding references and supplemented our related work section. The citations for these articles are numbered as follows: 18, 19, 21-25.